# Downregulation of Protease Cathepsin D and Upregulation of Pathologic α-Synuclein Mediate Paucity of DNAJC6-Induced Degeneration of Dopaminergic Neurons

**DOI:** 10.3390/ijms25126711

**Published:** 2024-06-18

**Authors:** Ching-Chi Chiu, Ying-Ling Chen, Yi-Hsin Weng, Shu-Yu Liu, Hon-Lun Li, Tu-Hsueh Yeh, Hung-Li Wang

**Affiliations:** 1Department of Medical Biotechnology and Laboratory Science, College of Medicine, Chang Gung University, Taoyuan 33302, Taiwan; ccchei@mail.cgu.edu.tw; 2Neuroscience Research Center, Chang Gung Memorial Hospital at Linkou, Taoyuan 33305, Taiwan; yhweng@cgmh.org.tw; 3Healthy Aging Research Center, College of Medicine, Chang Gung University, Taoyuan 33302, Taiwan; 4Department of Nursing, Chang Gung University of Science and Technology, Taoyuan 33303, Taiwan; ylchen@gw.cgust.edu.tw; 5Division of Movement Disorders, Department of Neurology, Chang Gung Memorial Hospital at Linkou, Taoyuan 33305, Taiwan; 6College of Medicine, Chang Gung University, Taoyuan 33302, Taiwan; 7Department of Physiology and Pharmacology, College of Medicine, Chang Gung University, No. 259, Wen-Hwa 1st Road, Kweishan, Taoyuan 33302, Taiwan; a0934188711@gmail.com; 8Department of Anesthesiology, Chang Gung Memorial Hospital at Linkou, Taoyuan 33305, Taiwan; h4567980@gmail.com; 9Department of Neurology, Taipei Medical University Hospital, Taipei 11031, Taiwan

**Keywords:** PARK19, DNAJC6, dopaminergic neurons, cathepsin D, α-synuclein

## Abstract

A homozygous mutation of the *DNAJC6* gene causes autosomal recessive familial type 19 of Parkinson’s disease (PARK19). To test the hypothesis that PARK19 DNAJC6 mutations induce the neurodegeneration of dopaminergic cells by reducing the protein expression of functional DNAJC6 and causing DNAJC6 paucity, an in vitro PARK19 model was constructed by using shRNA-mediated gene silencing of endogenous DANJC6 in differentiated human SH-SY5Y dopaminergic neurons. shRNA targeting DNAJC6 induced the neurodegeneration of dopaminergic cells. DNAJC6 paucity reduced the level of cytosolic clathrin heavy chain and the number of lysosomes in dopaminergic neurons. A DNAJC6 paucity-induced reduction in the lysosomal number downregulated the protein level of lysosomal protease cathepsin D and impaired macroautophagy, resulting in the upregulation of pathologic α-synuclein or phospho-α-synuclein^Ser129^ in the endoplasmic reticulum (ER) and mitochondria. The expression of α-synuclein shRNA or cathepsin D blocked the DNAJC6 deficiency-evoked degeneration of dopaminergic cells. An increase in ER α-synuclein or phospho-α-synuclein^Ser129^ caused by DNAJC6 paucity activated ER stress, the unfolded protein response and ER stress-triggered apoptotic signaling. The lack of DNAJC6-induced upregulation of mitochondrial α-synuclein depolarized the mitochondrial membrane potential and elevated the mitochondrial level of superoxide. The DNAJC6 paucity-evoked ER stress-related apoptotic cascade, mitochondrial malfunction and oxidative stress induced the degeneration of dopaminergic neurons via activating mitochondrial pro-apoptotic signaling. In contrast with the neuroprotective function of WT DNAJC6, the PARK19 DNAJC6 mutants (Q789X or R927G) failed to attenuate the tunicamycin- or rotenone-induced upregulation of pathologic α-synuclein and stimulation of apoptotic signaling. Our data suggest that PARK19 mutation-induced DNAJC6 paucity causes the degeneration of dopaminergic neurons via downregulating protease cathepsin D and upregulating neurotoxic α-synuclein. Our results also indicate that PARK19 mutation (Q789X or R927G) impairs the DNAJC6-mediated neuroprotective function.

## 1. Introduction

Patients with familial type 19 of Parkinson’s disease (PARK19) display early- or juvenile-onset parkinsonism phenotypes and autosomal recessive inheritance [1,2]. Homozygous mutations of the DNAJC6 gene, which encodes neuronal protein auxilin, are the genetic cause of PARK19 in different ethnic populations [1,2,3,4,5,6,7]. DNAJC6 (or auxilin) is a member of the family of J-domain proteins (JDPs, also referred as DNAJs or HSP40s), which act as crucial functional partners (or co-chaperones) of 70-kDa heat shock protein (HSP70) chaperones by enhancing the ATPase activity of HSP70 and playing an essential role in the initial protein substrate recognition of HSP70 [8,9,10]. DNAJC6 is highly expressed in CNS neurons [11] and functions as a neuronal co-chaperone of heat shock cognate 70 (HSC70)/HSPA8 [12], which is a major member of the HSP70 chaperone family [13]. Juvenile-onset PARK19 patients possess a homozygous (c.801-2A>G) mutation within intron 6 of the DNAJC6 gene, which causes defective splicing of DNAJC6 mRNA and the absence of normal DNAJC6 mRNA [3]. Homozygous nonsense truncating mutations (R256X, Q734X, Q789X and R806X) of DNAJC6 were observed in juvenile-onset PARK19 patients [4,5,6]. A novel homozygous mutation (R927G) of DNAJC6 was present in early-onset PARK19 patients [7]. It is generally believed that autosomal recessive PARK19 results from an intronic mutation of the DNAJC6 gene, which impairs the splicing of DNAJC6 mRNA, and a truncating or missense mutation of DNAJC6, which causes the absence of full-length functional DNAJC6, leading to a loss of the DNAJC6-mediated neuroprotective effect in dopaminergic neurons [14]. Further studies are required to elucidate the molecular pathomechanisms underlying the DNAJC6 paucity-induced neurodegeneration of dopaminergic cells and prove that a PARK19 truncating or missense mutation of DNAJC6 impairs the neuroprotective function of DNAJC6.

The typical neuropathological characteristics of Parkinson’s disease (PD) is the aggregation of α-synuclein and phospho-α-synuclein within Lewy bodies and Lewy neurites of surviving neurons [15,16]. Autosomal dominant PD is caused by the genetic amplification of α-synuclein and resulting increased protein expression of α-synuclein [17,18]. The neuronal death of SN dopaminergic cells is induced following the exogenous application of WT human α-synuclein into rat substantia nigra (SN) [19]. Intrastriatal application of misfolded α-synuclein caused the interneuronal transmission of neurotoxic α-synuclein and degeneration of SN dopaminergic neurons and PD motor deficits [20]. Therefore, the upregulated expression of pathologic α-synuclein is one of the primary pathogenic mechanisms underlying the cell death of SN dopaminergic neurons observed in PD patients [15,21,22].

Intraneuronal α-synuclein is mainly catabolized by two autophagy–lysosome routes, macroautophagy and chaperone-mediated autophagy (CMA) [23,24,25,26]. Macroautophagy requires the fusion of autophagosomes and lysosomes and the resulting formation of autolysosomes, which then carry out the lysosomal hydrolase-mediated degradation of sequestered materials. For the process of CMA, protein substrates, including α-synuclein with a KFERQ-like pentapeptide motif, are recognized by the HSC70 chaperone. Then, the protein substrate–HSC70 complex binds to LAMP2a, which mediates the translocation of protein substrates into lysosomes for subsequent degradation. The inhibition of either macroautophagy or CMA has been shown to upregulate the protein level of pathological α-synuclein within neurons [23,25,26,27,28]. The macroautophagy- or CMA-mediated degradation of intraneuronal α-synuclein requires a normal number and function of lysosomes. The physiological importance of lysosomal function in regulating the intracellular level of α-synuclein is indicated by the finding that genetic mutations of several lysosomal proteins, including GBA1, ATP13A2 and TMEM175, are believed to cause the upregulation of neurotoxic α-synuclein and resulting PD [29,30]. Therefore, a lysosomal deficiency-induced impairment of macroautophagy or CMA could cause the overexpression of pathologic α-synuclein and subsequent neuronal death of dopaminergic cells.

At the terminal step of autophagy, functional lysosomes are regenerated by the procedure of autophagic lysosome reformation (ALR), which is needed to maintain the number and homeostasis of functional lysosomes [31,32,33,34]. ALR consists of four steps: (1) Clathrin induces the creation of PtdIns(4,5)P2-containing microdomains on autolysosomes, which recruit the motor protein KIF5B. (2) KIF5B drives tubule formation from autolysosomes. (3) DNM2/dynamin 2-mediated scission at the tip of tubule generates protolysosomes. (4) Protolysosomes mature into functional lysosomes [32,34,35,36]. Knocking down the protein expression of clathrin heavy chain impairs the initiation of ALR [35], indicating that a normal intracellular level of free clathrin is required for ALR-mediated lysosomal homeostasis. One of the well-known physiological functions mediated by the HSC70-DNAJC6 complex in the nervous system is clathrin uncoating of the clathrin-coated vesicles, which is an essential step during the clathrin-mediated endocytosis process [11,37]. DNAJC6 is recruited to clathrin-coated vesicles. Subsequently, the J-domain of DNAJC6 recruits HSC70 and stimulates the ATPase activity of HSC70, which then causes the uncoating and disassembly of clathrin coats [37]. Therefore, DNAJC6 plays a critical role in regulating the homeostasis of intracellular free clathrin. With the paucity of functional DNAJC6 resulting from PARK19 mutations, a defective HSC70-DNAJC6 complex-mediated clathrin-uncoating process could cause a reduction in the intraneuronal level of free clathrin and the resulting impairment of ALR. Impaired ALR is expected to cause dysregulated lysosomal homeostasis and decrease the number of lysosomes, leading to a malfunction in the macroautophagy- or CMA-mediated clearance of neurotoxic α-synuclein.

Cathepsin D is a major lysosomal aspartic protease and functions as the main lysosomal hydrolase responsible for the macroautophagy- or CMA-mediated degradation of α-synuclein [38,39]. Lysosomes obtained from cathepsin D-deficient cells displayed a significantly impaired activity of degrading α-synuclein [40]. The protein level of α-synuclein or phospho-α-synuclein^Ser129^, the major pathological form of α-synuclein [41], was significantly increased in the brain of a cathepsin D-deficient mouse [42,43]. The overexpression of recombinant cathepsin D decreased the protein level of α-synuclein in a cathepsin D knockout mouse and the iPSC-derived dopaminergic neurons of a PD patient carrying A53T, a mutant α-synuclein [44]. Interestingly, cathepsin D has been shown to be a candidate PD susceptibility gene [45]. A DNAJC6 deficiency of PARK19-induced ALR impairment and the resulting reduction in lysosomal number could downregulate the protein expression of lysosomal cathepsin D, leading to the impaired macroautophagy- or CMA-mediated clearance of α-synuclein and subsequent upregulation of pathologic α-synuclein.

α-Synuclein and phospho-α-synuclein^Ser129^ are found within the endoplasmic reticulum (ER) [46], and increased ER α-synuclein or phospho-α-synuclein^Ser129^ activates ER stress and the unfolded protein response (UPR) [47,48]. In the presence of constant ER stress, the UPR triggers the apoptotic cascade and induces neuronal death [48,49]. The neuronal death of SN dopaminergic cells caused by an ER stress-evoked apoptotic pathway is implicated in the etiopathogenesis of PD [50,51]. α-Synuclein is also located within the mitochondria [52,53]. Upregulated mitochondrial α-synuclein causes dysfunction and oxidative stress in the mitochondria [52,53,54] and the subsequent induction of mitochondrial apoptotic signaling, which then promotes the neurodegeneration of SN dopaminergic cells [55,56]. It is very likely that DNAJC6 paucity of PARK19-induced downregulation of protease cathepsin D increases ER or mitochondrial α-synuclein and phospho-α-synuclein^Ser129^, resulting in the stimulation of ER stress-triggered or mitochondrial apoptotic signaling and subsequent neuronal death of SN dopaminergic cells.

In this study, it was hypothesized that PARK19 loss-of-function mutations of DNAJC6 promote the cell death of dopaminergic neurons via inducing a defective expression of functional DNAJC6. We investigated the molecular pathomechanisms underlying the DNAJC6 paucity-induced degeneration of dopaminergic neurons by using a PARK19 cellular model, which was constructed by knocking down endogenous DANJC6 expression in differentiated human SH-SY5Y dopaminergic neurons. Our results suggest that a DNAJC6 deficiency decreases the number of lysosomes and protein level of lysosomal protease cathepsin D and causes macroautophagy impairment, leading to the upregulation of ER and mitochondrial α-synuclein or phospho-α-synuclein^Ser129^ and subsequent induction of ER stress-evoked and mitochondrial apoptotic signaling. Our data also demonstrate that in contrast with the neuroprotective function of WT DNAJC6, PARK19 DNAJC6 mutants (Q789X or R927G) fail to suppress the tunicamycin- or rotenone-evoked upregulation of pathologic α-synuclein and stimulation of the apoptotic pathway within dopaminergic neurons.

## 2. Results

### 2.1. WT DNAJC6 Exerts a Neuroprotective Effect on Dopaminergic Neurons

We hypothesized that homozygous exonic or intronic genetic mutations of DNAJC6 cause the absence of functional DNAJC6 and a loss of the DNAJC6-mediated neuroprotective function, resulting in the neuronal death of SN dopaminergic cells and PARK19. We prepared a cellular model of PARK19 by reducing endogenous DNAJC6 protein in differentiated human SH-SY5Y dopaminergic neurons [57,58,59] using shRNAs of DNAJC6. A three-day transfection of human DNAJC6 shRNAs led to an about 90% decrease in the protein expression of endogenous DNAJC6 (Figure 1A). The impaired expression of functional DNAJC6 caused by a four-day transfection of shRNAs targeting DNAJC6 induced significant neuronal death of dopaminergic cells (Figure 1B). The exogenous expression of WT human DNAJC6 prevented the DNAJC6 shRNA-induced degeneration of dopaminergic neurons (Figure 1B). Our finding demonstrates that WT DNAJC6 supports the viability of dopaminergic neurons by possessing a neuroprotective function.

### 2.2. DNAJC6 Deficiency Reduces Protein Level of Intraneuronal Clathrin Heavy Chain and Number of Lysosomes in Dopaminergic Neurons

The HSC70-DNAJC6 complex causes the uncoating and disassembly of clathrin coats from clathrin-coated vesicles during clathrin-mediated endocytosis [11,37] and plays a critical role in regulating the homeostasis of intraneuronal free clathrin. A PARK19 mutation-induced deficiency of functional DNAJC6 could impair the HSC70-DNAJC6 complex-mediated clathrin-uncoating process and cause the dysregulated expression of intracellular clathrin. In line with this hypothesis, a DNAJC6 shRNA-induced DNAJC6 deficiency significantly decreased the cytosolic protein level of clathrin heavy chain in dopaminergic neurons (Figure 2).

Clathrin heavy chain plays a crucial role in initiating the autophagic lysosome reformation (ALR) process [31,32,33,34,35], which mediates the regeneration of functional lysosomes at the terminal stage of autophagy and is required for maintaining a normal number of lysosomes. Thus, the DNAJC6 deficiency-induced downregulation of cytosolic clathrin heavy chain (Figure 2) could cause the impairment of ALR and the resulting decreased number of lysosomes in dopaminergic neurons. To test this hypothesis, lysosomes within dopaminergic neurons were visualized by conducting live cell imaging of lysosomes using LysoTracker, which specifically stains acidic lysosomes [60], and immunofluorescence staining of LAMP2, which is a major lysosomal marker protein [61]. DNAJC6 paucity caused by a 3-day transfection of shRNAs targeting DNAJC6 decreased the fluorescence intensity of LysoTracker Yellow in dopaminergic neurons (Figure 3A). A three-day transfection of DNAJC6 shRNA also reduced the fluorescence intensity of LAMP2-positive lysosomes in dopaminergic neurons (Figure 3B). Immunoblotting analysis further demonstrated that the cytosolic protein levels of LAMP1 and LAMP2, which are membrane protein markers of lysosomes, were decreased in DNAJC6 shRNA-transfected dopaminergic neurons (Figure 2). These results suggest that a PARK19 mutation-induced DNAJC6 deficiency decreases the number of lysosomes within dopaminergic neurons.

### 2.3. Paucity of DNAJC6 Causes Macroautophagy Impairment and Decreases Protein Level of Protease Cathepsin D, Resulting in Upregulation of Pathologic α-Synuclein or Phospho-α-Synuclein^ser129^ within Dopaminergic Neurons

Macroautophagy mediates the degradation of autophagy receptor protein p62/SQSTM1 (sequestosome-1) [62]; macroautophagy impairment causes p62/SQSTM1 accumulation. Therefore, an upregulated protein level of p62/SQSTM1 indicates defective macroautophagy [62]. Macroautophagy activity requires a normal number of lysosomes. A DNAJC6 deficiency-induced reduction in the lysosomal number (Figure 3) is likely to cause impaired macroautophagy. In line with this hypothesis, an upregulation of cytosolic p62/SQSTM1, which indicates impaired macroautophagy, was observed in DNAJC6 shRNA-transfected dopaminergic neurons (Figure 2).

Cathepsin D is a major lysosomal aspartic protease and is mainly responsible for the macroautophagy- or CMA-mediated lysosomal degradation of α-synuclein [38,39]. A deficiency of functional DNAJC6 caused by PARK19 mutations could decrease the protein level of lysosomal protease cathepsin D within dopaminergic neurons by reducing the number of lysosomes (Figure 3). In accordance with this hypothesis, DNAJC6 paucity induced by a 3-day transfection of DNAJC6 shRNAs significantly reduced the cytosolic protein level of protease cathepsin D in dopaminergic neurons (Figure 2).

The inhibition of macroautophagy upregulates the protein level of intraneuronal α-synuclein [23,25,26,27,28]. A knockdown of lysosomal protease cathepsin D impaired the degradation of intracellular α-synuclein [40], and a knockout of cathepsin D caused an upregulation of pathologic α-synuclein or phospho-α-synuclein^Ser129^ in the brain [42,43]. Thus, DNAJC6 deficiency-induced macroautophagy impairment and the downregulation of protease cathepsin D could upregulate neurotoxic α-synuclein and phospho-α-synuclein^Ser129^ within dopaminergic neurons. Immunoblotting assays using cellular lysates showed that the intraneuronal α-synuclein or phospho-α-synuclein^Ser129^ level was significantly increased in DNAJC6 shRNA-transfected dopaminergic neurons (Figure 4). This finding suggests that a PARK19 mutation-induced DNAJC6 deficiency impairs macroautophagy and downregulates protease cathepsin D, leading to the upregulation of neurotoxic α-synuclein or phospho-α-synuclein^Ser129^ and subsequent degeneration of dopaminergic neurons.

### 2.4. shRNA Targeting α-Synuclein or Overexpression of FLAG-Tagged Cathepsin D Blocks DNAJC6 Deficiency-Induced Degeneration of Dopaminergic Cells

We hypothesized that a PARK19 mutation-induced DNAJC6 deficiency induces the neurodegeneration of dopaminergic cells by upregulating pathological α-synuclein. To test this hypothesis, shRNAs of α-synuclein, which knocked down about 85% of endogenous α-synuclein, were co-transfected with DNAJC6 shRNAs. In accordance with our hypothesis, the co-transfection of α-synuclein shRNA and resulting knockdown expression of neurotoxic α-synuclein significantly inhibited the DNAJC6 shRNA-induced neurodegeneration of dopaminergic cells (Figure 5A).

We also hypothesized that a deficiency in the DNAJC6-induced downregulation of protease cathepsin D upregulates the protein expression of neurotoxic α-synuclein, resulting in the neuronal death of dopaminergic cells. This hypothesis is supported by the finding that the overexpression of FALG-tagged cathepsin D significantly prevented DNAJC6 shRNA-triggered degeneration of dopaminergic cells (Figure 5B). Following the overexpression of FLAG-tagged cathepsin D, shRNA1 or shRNA2 of DNAJC6 also did not significantly increase the protein expression of intraneuronal α-synuclein (*n* = 4 experiments). A future study using the PARK19 mouse model is required to provide in vivo evidence that PARK19 mutation-induced DNAJC6 paucity causes the degeneration of SN dopaminergic neurons via downregulating protease cathepsin D and upregulating pathologic α-synuclein.

### 2.5. Lack of DNAJC6 Upregulates ER α-Synuclein or Phospho-α-Synuclein^ser129^ and Activates ER Stress, UPR and ER Stress-Triggered Pro-Apoptotic Signaling within Dopaminergic Neurons

α-Synuclein and phospho-α-synuclein^Ser129^ are found within the ER [46], and increased ER α-synuclein or phospho-α-synuclein^Ser129^ activates ER stress and the UPR [47,48]. The DNAJC6 deficiency-induced upregulation of intraneuronal α-synuclein or phospho-α-synuclein^Ser129^ (Figure 4) should increase ER α-synuclein or phospho-α-synuclein^Ser129^ and activate ER stress and the UPR. Immunoblotting assays showed that the protein level of ER α-synuclein or phospho-α -synuclein^Ser129^ was increased in DNAJC6 shRNA-transfected dopaminergic neurons (Figure 6A). Immunoblotting analysis further indicated that protein levels of ER chaperones, including calnexin, Ero1-Lα, Grp78 and PDI, and UPR markers, including IRE1α, PERK, phospho-IRE1α^Ser724^ and phospho-PERK^Thr980^, were significantly upregulated in DNAJC6 shRNA-transfected dopaminergic neurons (Figure 6B).

In the presence of incessant ER stress, the UPR activates the apoptotic cascade and causes neurodegeneration [48,49]. Upregulated phospho-PERK augments the mRNA and protein levels of ATF4, which then stimulates the transcription of pro-apoptotic Noxa and CHOP, which enhances the gene transcription of pro-apoptotic Bim and Puma [48,49]. Persistent ER stress also activates caspase-12, which then activates downstream caspase-9 [63]. A deficiency in the DNAJC6-induced stimulation of ER stress and the UPR could cause the neurodegeneration of dopaminergic cells by activating ER stress-evoked apoptotic signaling. In line with this hypothesis, mRNA expressions of ATF4, Bim, CHOP, Puma and Noxa (Figure 7A) and cytosolic protein expressions of ATF4, Bim, CHOP, Puma, Noxa and active caspase-12 (Figure 7B) were upregulated in DNAJC6 shRNA-transfected dopaminergic neurons.

### 2.6. Paucity of DNAJC6 Increases Mitochondrial α-Synuclein or Phospho-α-Synuclein^ser129^ and Induces Mitochondrial Impairment and Oxidative Stress in Dopaminergic Neurons

α-Synuclein is localized in the mitochondria [52,53], and upregulated mitochondrial α-synuclein induces malfunction and oxidative stress in the mitochondria [52,53,54]. The DNAJC6 deficiency-induced upregulation of intraneuronal α-synuclein or phospho-α-synuclein^Ser129^ (Figure 4) should increase mitochondrial α-synuclein or phospho-α-synuclein^Ser129^, resulting in mitochondrial impairment and oxidative stress. In accordance with this hypothesis, an upregulated protein level of mitochondrial α-synuclein or phospho-α-synuclein^Ser129^ was observed in DNAJC6 shRNA-transfected dopaminergic neurons (Figure 8A). Live cell imaging analysis demonstrated that a 3-day transfection of DNAJC6 shRNAs reduced the fluorescence signal of mitochondrial membrane potential (ΔΨm) indicator TMRM and depolarized ΔΨm in dopaminergic neurons (Figure 8B). An increased fluorescence signal of mitochondrial superoxide dye MitoSox Red and upregulated mitochondrial level of superoxide, which is the major ROS, were also observed in DNAJC6 shRNA-transfected dopaminergic neurons (Figure 8C).

### 2.7. DNAJC6 Paucity Promotes Apoptotic Death of Dopaminergic Neurons via Activating Mitochondrial Apoptotic Signaling

The upregulation of Bim, Puma or Noxa stimulates the mitochondria-mediated apoptotic pathway via enhancing the translocation of mitochondrial cytochrome c to cytoplasm and activating caspase-9 and caspase-3 [64]. An increased mitochondrial ROS level and the depolarization of Ψm triggers the activation of mitochondrial pro-apoptotic signaling via augmenting the release of mitochondrial cytochrome c [55,56]. In the present study, a DNAJC6 deficiency is likely to activate mitochondria-mediated apoptotic signaling and induce the apoptotic death of dopaminergic neurons via increasing the expression of Bim, Puma or Noxa (Figure 7) and causing depolarized ΔΨm and the upregulation of mitochondrial ROS (Figure 8B,C). Cytochrome c was mainly found within the mitochondria of control or SC shRNA-transfected dopaminergic cells (Figure 9A). The DNAJC6 deficiency induced by a 3-day transfection of DNAJC6 shRNA significantly decreased mitochondrial cytochrome c and elevated cytosolic cytochrome c in dopaminergic neurons (Figure 9A). Upregulated cytosolic active caspase-9 and active caspase-3 were also found in DNAJC6 shRNA-transfected dopaminergic neurons (Figure 9A). A four-day transfection of shRNAs targeting DNAJC6 also significantly increased the percentage of TUNEL-positive dopaminergic neurons (Figure 9B).

### 2.8. PARK19 DNAJC6 Mutant Fails to Reverse Tunicamycin- or Rotenone-Evoked Upregulation of Pathologic α-Synuclein and Stimulation of Apoptotic Pathway in Dopaminergic Neurons

To test the hypothesis that PARK19 mutation (Q789X or R927G) causes the loss of the DNAJC6-induced neuroprotective effect and the resulting degeneration of dopaminergic neurons, FLAG-tagged WT, Q789X or R927G human DNAJC6 was transiently expressed in differentiated SH-SY5Y dopaminergic neurons (Figure 10A). Then, we investigated the protective function of WT or PARK19 DNAJC6 on neurotoxicity induced by tunicamycin and rotenone, which cause neuronal death by increasing neurotoxic α-synuclein and activating pro-apoptotic signaling [65,66]. Neurotoxin tunicamycin (TCM; 1 μM) caused the degeneration of dopaminergic neurons (Figure 10B) by upregulating the cytosolic expression of pathological α-synuclein and proteins implicated in the ER stress-evoked pro-apoptotic cascade, including active caspase-12, CHOP, Grp78, IRE1α and PERK (Figure 10C). WT DNAJC6 inhibited the tunicamycin-evoked neuronal death of dopaminergic cells and upregulation of cytosolic α-synuclein, active caspase-12, CHOP, Grp78, IRE1α or PERK (Figure 10B,C). In contrast, the PARK19 DNAJC6 mutants (Q789X or R927G) did not reverse the tunicamycin-induced degeneration of dopaminergic neurons and upregulation of cytosolic α-synuclein, active caspase-12, CHOP, Grp78, IRE1α or PERK (Figure 10B,C). Treating dopaminergic neurons with neurotoxin rotenone (RTN; 0.5 μM) reduced the neuronal viability (Figure 10B) by upregulating cytosolic pathologic α-synuclein and inducing the stimulation of the mitochondria-mediated apoptotic cascade, which is shown by increased cytosolic cytochrome c, active caspase-9 or active caspase-3 levels (Figure 10D). WT DNAJC6 prevented the rotenone-triggered degeneration of dopaminergic neurons and upregulation of cytosolic α-synuclein, cytochrome c, active caspase-9 and active caspase-3 (Figure 10B,D). The PARK19 DNAJC6 mutants (Q789X or R927G) did not attenuate the rotenone-induced degeneration of dopaminergic neurons and cytosolic upregulation of α-synuclein, cytochrome c, active caspase-9 or active caspase-3 (Figure 10B,D). These results provide the evidence that the PARK19 DNAJC6 mutants (Q789X or R927G) cause the neurodegeneration of dopaminergic cells by impairing the neuroprotective effect of DNAJC6.

## 3. Discussion

Molecular genetic studies identified homozygous mutations of the DNAJC6 gene as the cause of autosomal recessive PARK19 [1,2,3,4,5,6,7]. In this study, it was hypothesized that homozygous exonic or intronic mutations of the DNAJC6 gene cause a deficiency of functional DNAJC6 and the lack of a DNAJC6-mediated neuroprotective effect, resulting in the neuronal death of SN dopaminergic cells and PARK19. To investigate this hypothesis, a PARK19 cellular model was prepared with the aid of the shRNA-induced knockdown of endogenous DNAJC6 in differentiated human SH-SY5Y dopaminergic neurons and utilized to study the molecular pathomechanisms underlying the DNAJC6 paucity-evoked neurodegeneration of dopaminergic cells. Consistent with our hypothesis, DNAJC6 paucity caused by shRNAs of DNAJC6 led to the degeneration of dopaminergic neurons, indicating that WT DNAJC6 possesses a neuroprotective function and supports the viability of dopaminergic neurons.

The upregulated expression and accumulation of neurotoxic α-synuclein is one of the major pathomechanisms underlying PD-related neuronal death of SN dopaminergic cells [15,21,22]. Two autophagy–lysosome pathways, macroautophagy and chaperone-mediated autophagy (CMA), carry out the degradation of intracellular α-synuclein [23,24,25,26]. A normal number and function of lysosomes is required for the macroautophagy- or CMA-mediated breakdown of pathological α-synuclein. A lysosomal deficiency-induced impairment of macroautophagy or CMA is believed to trigger the cell death of dopaminergic neurons in PD by increasing the intraneuronal expression of pathogenic α-synuclein [29,30]. Autophagic lysosome reformation (ALR) mediates the regeneration of functional lysosomes at the terminal step of autophagy and plays a vital role in maintaining a normal number and the homeostasis of lysosomes [31,32,33,34]. Clathrin is involved in the initiation of ALR, and decreased clathrin heavy chain impairs ALR-mediated lysosomal homeostasis [32,34,35]. The HSC70-DNAJC6 complex initiates the uncoating and disassembly of clathrin coats from clathrin-coated vesicles during clathrin-mediated endocytosis [11,37] and plays a crucial role in controlling the homeostasis of intraneuronal free clathrin. A PARK19 mutation-induced deficiency of functional DNAJC6 could damage the HSC70-DNAJC6 complex-mediated clathrin-uncoating process and dysregulate the protein expression of intraneuronal clathrin. Consistent with this hypothesis, the cytosolic level of clathrin heavy chain was reduced in DNAJC6 shRNA-transfected dopaminergic neurons. A DNAJC6 deficiency-induced reduction in the clathrin heavy chain level is expected to impair the initiation and function of ALR, leading to a reduction in the number of intraneuronal lysosomes. Consistent with this hypothesis, imaging of lysosomes using LysoTracker staining and immunofluorescence staining of lysosomal marker LAMP2 demonstrated that the transfection of DNAJC6 shRNA reduced the number of lysosomes within dopaminergic neurons. A DNAJC6 paucity-induced reduction in the lysosomal number is further supported by the finding that cytosolic protein levels of LAMP1 and LAMP2 were decreased in DNAJC6 shRNA-transfected dopaminergic neurons. Our results suggest that a PARK19 mutation-induced DNAJC6 deficiency causes a defective lysosomal function in dopaminergic neurons by reducing the cytosolic protein level of clathrin heavy chain, impairing the ALR process and decreasing the number of lysosomes. In addition to PARK19, several hereditary neurodegenerative diseases have also been shown to result from an ALR impairment-induced reduction in the lysosomal number within neurons. Spatacsin and spastizin play critical roles in initiating the tubule formation of autolysosomes during the ALR process [67]. Loss-of-function mutations of spatacsin and spastizin, which lead to the impairment of ALR and depletion of free lysosomes [67,68], cause autosomal recessive spastic paraplegia type 11 and spastic paraplegia type 15, respectively [34,67,68]. Homozygous mutations of lysosomal trafficking regulator (LYST) cause autosomal recessive Chediak–Higashi syndrome with progressive neurodegeneration by impairing the scission of autolysosome tubules during ALR and reducing the number of neuronal lysosomes [69]. Loss-of-function mutations of CLN3 cause Batten disease, which is one of the most devastating childhood neurodegenerative diseases, by suppressing the tubule formation of autolysosomes and impairing ALR [70].

Macroautophagy-mediated degradation of neurotoxic α-synuclein within dopaminergic neurons requires a normal number of lysosomes. A DNAJC6 deficiency-induced decrease in the number of lysosomes caused an impairment of macroautophagy, which is indicated by the upregulated protein expression of cytosolic p62/SQSTM1 [62] in DNAJC6 shRNA-transfected dopaminergic neurons. A knockout of lysosomal aspartic protease cathepsin D upregulates the protein expression of α-synuclein in the brain [42,43], and the overexpression of cathepsin D decreases the protein level of intraneuronal α-synuclein [44]. Therefore, lysosomal protease cathepsin D plays a major role in the macroautophagy- or CMA-mediated clearance of α-synuclein [38,39]. A DNAJC6 deficiency-induced decrease in the number of lysosomes could decrease the protein level of lysosomal cathepsin D within dopaminergic neurons. Consistent with our hypothesis, the cytosolic protease cathepsin D level was downregulated in DNAJC6 shRNA-transfected dopaminergic neurons. A PARK19 mutation-induced DNAJC6 deficiency could cause macroautophagy impairment and the downregulation of lysosomal protease cathepsin D, leading to an upregulated level of neurotoxic α-synuclein or phospho-α-synuclein^Ser129^ and the resulting neurodegeneration of SN dopaminergic cells. In accordance with this hypothesis, the intracellular protein level of pathologic α-synuclein or phospho-α-synuclein^Ser129^ was increased in DNAJC6 shRNA-transfected dopaminergic neurons. shRNA targeting α-synuclein also inhibited the DNAJC6 paucity-induced neurodegeneration of dopaminergic cells. Our hypothesis is further supported by the finding that the overexpression of FLAG-tagged cathepsin D significantly prevented the DNAJC6 shRNA-induced upregulation of intraneuronal α-synuclein and neuronal death of dopaminergic cells. According to our results, it is reasonable to hypothesize that drugs, which activate macroautophagy or cathepsin D, could possess therapeutic effects on PARK19 degeneration of SN dopaminergic neurons. Interestingly, a reduced activity and protein level of cathepsin D were found in the CSF and plasma of sporadic PD patients, respectively [71,72]. A decreased activity or protein level of cathepsin D was also found in the SN of patients with sporadic PD [73,74]. Therefore, a reduced level of protease cathepsin D and the resulting upregulation of neurotoxic α-synuclein is likely to be implicated in the etiopathogenesis of both PARK19 and sporadic PD.

The DNAJC6 deficiency-induced downregulation of protease cathepsin D increased ER α-synuclein or phospho-α-synuclein^Ser129^ in dopaminergic neurons. In accordance with previous studies reporting that the upregulation of ER α-synuclein or phospho-α-synuclein^Ser129^ activates ER stress and the UPR [47,48], the presence of ER stress, which is demonstrated by increased protein levels of cytosolic calnexin, Ero1-Lα, Grp78 and PDI, and the UPR, which is indicated by upregulated protein expressions of cytosolic IRE1α, PERK, phospho-IRE1α^Ser724^ and phospho-PERK^Thr980^, was found in DNAJC6 shRNA-transfected dopaminergic neurons. We hypothesized that a DNAJC6 deficiency from PARK19 mutation induces the apoptotic death of SN dopaminergic cells via the ER stress-triggered activation of apoptotic signaling and caspase-12. In line with this hypothesis, a deficiency of DNAJC6 upregulated the mRNA or protein expressions of Bim, Noxa, CHOP, ATF4, Puma and active caspase-12 in dopaminergic neurons. In addition to DNAJC6, our recent study showed that a loss or mutation of RAB39B gene-induced RAB39B paucity, which causes X-chromosome-linked PD, also increases ER α-synuclein or phospho-α-synuclein^Ser129^ and triggers the neurodegeneration of dopaminergic cells via stimulating the ER stress-evoked apoptotic pathway [75].

The paucity of the DNAJC6-mediated reduction of protease cathepsin D upregulated mitochondrial α-synuclein or phospho-α-synuclein^Ser129^ in dopaminergic neurons. The upregulation of mitochondrial α-synuclein and phospho-α-synuclein^Ser129^ caused malfunction and oxidative stress in the mitochondria, which is demonstrated by the depolarized mitochondrial membrane potential and increased superoxide level within the mitochondria of DNAJC6 shRNA-transfected dopaminergic neurons. Impairment and oxidative stress in the mitochondria cause the neuronal death of SN dopaminergic cells and is implicated in the pathogenesis of idiopathic and familial PD [55,56,76]. Our data suggest that the paucity of DNAJC6-induced mitochondrial impairment and oxidative stress is also implicated in the molecular etiopathogenesis of PARK19. The DNAJC6 deficiency-induced upregulation of Noxa, Bim or Puma and mitochondrial malfunction or oxidative stress could trigger mitochondrial pro-apoptotic signaling via augmenting the translocation of mitochondrial cytochrome c to the cytoplasm and inducing the activation of caspase-9 and caspase-3, resulting in the apoptotic death of SN dopaminergic cells [55,56,64]. In line with this hypothesis, the cytosolic upregulation of cytochrome c, active caspase-9 and active caspase-3 was found in DNAJC6 shRNA-transfected dopaminergic neurons. DNAJC6 paucity also significantly increased the percentage of TUNEL-positive cells in DNAJC6 shRNA-transfected dopaminergic neurons.

We hypothesized that the expression of WT DNAJC6 is essential for the survival of dopaminergic neurons by possessing a neuroprotective function and that PARK19 homozygous mutations of DNAJC6 (Q789X or R927G) cause a loss of the neuroprotective effect, resulting in the degeneration of SN dopaminergic neurons. We studied this hypothesis by evaluating the neuroprotective function of FLAG-tagged human WT or PARK19 DNAJC6 on the tunicamycin- or rotenone-evoked upregulation of pathogenic α-synuclein and stimulation of the ER stress-triggered or mitochondrial apoptotic cascade in dopaminergic neurons. In accordance with our hypothesis, WT DNAJC6 exerted a neuroprotective effect by preventing the tunicamycin- or rotenone-evoked upregulation of pathogenic α-synuclein and stimulation of ER stress-triggered or mitochondrial pro-apoptotic signaling. In contrast, the PARK19 DNAJC6 mutants (Q789X or R927G) failed to attenuate the upregulation of pathologic α-synuclein and activation of apoptotic pathway caused by tunicamycin or rotenone. C-terminal truncated (Q789X) or missense mutant (R927G) DNAJC6 does not exert a neuroprotective function, indicating that the C-terminal 789–970 amino acids of human DNAJC6, including the R927 residue, are needed for the DNAJC6-induced neuroprotective effect in dopaminergic neurons. Consistent with our result, the J-domain of human DNAJC6, which is required for interacting with HSC70 and HSC70-DNAJC6 complex-mediated physiological effects including neuroprotective function [8,9,10], is located at the C-terminal domain containing the R927 residue [14,77].

## 4. Materials and Methods

### 4.1. Cultured Differentiated SH-SY5Y Dopaminergic Neurons

Human SH-SY5Y neurons are tyrosine hydroxylase-positive and release dopamine [57,58]. SH-SY5Y dopaminergic cells were cultured with DMEM medium containing 10% FBS and the F-12 nutrient mixture. SH-SY5Y dopaminergic neurons was differentiated following a 3-day incubation of retinoic acid (10 μM) and decreased serum (3% FBS) medium [59].

### 4.2. Transfecting shRNAs to Differentiated SH-SY5Y Dopaminergic Neurons

The pLKO.1-puro vector containing shRNA of human DNAJC6 (shRNA1: 5′CCAGCTACACAAAGGGAGATT3′; shRNA2: 5′CGTGGGAAAGGATCAAGTAAT3′) was purchased from the National RNAi Core Facility at Academia Sinica, Taipei, Taiwan. The pRS shRNA vector containing shRNA targeting human α-synuclein (shRNA1: 5′TCAGAAGTTGTTAGTGATTTGCTATCATA3′; shRNA2: 5′GGTATCAAGACTACGAACCTGAAGCCTAA3′) was purchased from OriGene (Rockville, MD, USA). Scrambled control (SC) shRNA and DNAJC6 or α-synuclein shRNA were transfected into dopaminergic neurons with Lipofectamine 2000 transfection reagent (ThermoFisher, Waltham, MA, USA). Three or four days after transfection, various experiments, as described below, were performed using control or transfected neurons.

### 4.3. Construction of cDNA Encoding PARK19 DNAJC6

Oligonucleotide-directed PCR mutagenesis (Q5 Site Directed Mutagenesis Kit, NEB, Ipswich, MA, USA) was conducted to obtain the cDNA of PARK19 DNAJC6 mutants Q789X and R927G. Mutation primer 1 (5′CAACTGGCAGTAGCCACAGCCTA3′) and mutation primer 2 (5′GAAGGTGTACGGCAAGGCTGTCC3′) were used for the mutations of Q789X and R927G DNAJC6, respectively. PARK19 mutations of human DNAJC6 were verified using DNA sequencing, and cDNA encoding wild-type (WT), Q789X or R927G DNAJC6 was subcloned into a pcDNA3 expression vector containing a FLAG-tag sequence. Then, the cDNA of the FLAG-tagged WT or mutant DNAJC6 was transfected to differentiated SH-SY5Y dopaminergic neurons.

### 4.4. Measurement of Neuronal Viability

Differentiated human SH-SY5Y dopaminergic neurons plated into 6-well plate were transfected with SC shRNA or shRNA. Four days following the transfection, WST-8 was added into each well at 37 °C for 2 h. Then, the OD_450nm_ was measured using xMark spectrophotometer (Bio-Rad, Hercules, CA, USA).

The neuroprotective function of WT or PARK19 DNAJC6 was investigated by transfecting the cDNA of FLAG-tagged WT, Q789X or R927G DNAJC6 into differentiated SH-SY5Y dopaminergic cells. Subsequently, the dopaminergic neurons were treated with neurotoxic tunicamycin (1 μM) or rotenone (0.5 μM) for 3 days. Then, the OD_450nm_ value was obtained following a 2 h incubation of WST-8 at 37 °C.

### 4.5. Subcellularfractionation

According to our previous study [75], mitochondrial and cytosolic fractions were obtained from control or transfected dopaminergic neurons. Briefly, neurons were treated with lysis buffer containing 210 mM mannitol, 70 mM sucrose, 1 mM DTT, 1 mM EGTA, 10 mM HEPES (pH = 7.3), protease inhibitor cocktail (Sigma-Aldrich, St. Louis, MO, USA) and phosphatase inhibitor cocktail (Sigma-Aldrich). Dopaminergic neurons were then homogenized with a Dounce tissue grinder, and the lysate was centrifuged at 500× *g* for 10 min. The supernatant was centrifuged at 8100× *g* for 9 min, and the resulting pellet was obtained as mitochondrial fraction. Following a subsequent 20 min centrifugation at 14,000× *g*, the supernatant was used as cytosol fraction. The endoplasmic reticulum (ER) fraction was collected from dopaminergic neurons by using the Minute™ ER Enrichment Kit (Invent Biotechnologies, Plymouth, MN, USA).

### 4.6. Western Blot Analysis of Clathrin, p62/SQSTM1 and Lysosomal Proteins

Cytosolic protein extracts of dopaminergic neurons were obtained as described above and separated on 10% SDS-PAGE gels. PVDF membranes were then incubated with one of the following diluted primary antiserums purchased from Cell Signaling Technology: (1) rabbit monoclonal anti-clathrin heavy chain antibody; (2) rabbit anti-lysosomal-associated membrane protein 1 (LAMP1) monoclonal antiserum; (3) rabbit monoclonal anti-lysosomal-associated membrane protein 2 (LAMP2) antibody; (4) anti-p62/SQSTM1 rabbit monoclonal antiserum; (5) rabbit polyclonal anti-cathepsin D antibody. Following the interaction with an HRP-linked secondary antiserum, proteins on the membranes were visualized using an ECL reagent (Merck, Burlington, MA, USA). The protein bands were quantified and normalized to the actin signal.

### 4.7. Live Cell Staining of Lysosomes Using LysoTracker

LysoTracker is a cell permeable dye that stains acidic organelles and visualizes lysosomes in live cells [60]. Three days after transfecting the dopaminergic neurons with SC shRNA or DNAJC6 shRNA, the control or transfected dopaminergic neurons were incubated with 100 nM LysoTracker Yellow HCK-123 for 1 h at 37 °C. Then, a LionHeart FX automatic microscope (BioTek, Agilent, Santa Clara, CA) was utilized to capture 24 fields, which contain about800 cells, of LysoTracker Yellow fluorescent images. The fluorescence intensity was then measured and analyzed with Gen5 software (https://www.agilent.com/en/support/biotek-software-releases (accessed on 5 May 2024)) (BioTek). The LysoTracker signal of transfected dopaminergic neurons was compared with that of control neurons.

### 4.8. Immunofluorescence Staining of Lysosomal Marker Protein LAMP2

Immunofluorescence staining of LAMP2 was conducted to visualize the lysosomes of dopaminergic neurons [61]. Three days following transfection, control or transfected dopaminergic neurons plated into 6-well plate were fixed with paraformaldehyde and permeabilized with Triton X-100. Then, dopaminergic neurons were incubated at 4 °C overnight with diluted rabbit anti-LAMP2 monoclonal antibody (Cell Signaling Technology, Danvers, MA, USA) and interacted with Alexa Fluor 488-conjugated secondary antibody. Subsequently, a LionHeart FX automatic microscope (BioTek) was used to obtain fluorescent images of LAMP2-positive lysosomes from 24 fields, which contain about 800–850 neurons, per well. Fluorescent signals were examined with Gen5 software (BioTek), and the fluorescence intensity of transfected dopaminergic neurons was compared with that of control neurons.

### 4.9. Western Blot Analysis of α-Synuclein and Phospho-α-Synuclein^ser129^

Dopaminergic neurons were sonicated with lysis buffer containing 1% NP40, 0.5% sodium deoxycholate, 0.1% SDS, protease inhibitor cocktail (Sigma-Aldrich) and phosphatase inhibitor cocktail (Sigma-Aldrich). Following centrifugation at 13,000× *g* at 4 °C, the supernatant containing cellular lysate was used for immunoblotting assays. Proteins of the cellular lysate, ER fraction or mitochondrial fraction were separated on SDS-PAGE (15%) gels. Then, the membranes were interacted with anti-α-synuclein rabbit monoclonal antiserum (Abcam, Cambridge, UK) or anti-phospho-α-synuclein^Ser129^ rabbit monoclonal antibody (Abcam) at 4 °C overnight. Following the incubation with HRP-linked secondary antibody, immunoreactive proteins of membranes were detected with an ECL reagent (Merck).

### 4.10. Immunoblotting Assays of ER Stress, UPR or ER Stress-Evoked Pro-Apoptotic Cascade

Cytosolic protein extracts of control or transfected dopaminergic neurons were separated on SDS-PAGE (10% or 15%) gels. PVDF membranes were then incubated with the following primary antibodies: (1) rabbit anti-calnexin monoclonal antibody (Cell Signaling Technology); (2) rabbit polyclonal anti-Ero1-Lα antiserum (Cell Signaling Technology); (3) rabbit anti-Grp78 monoclonal antibody (Cell Signaling Technology); (4) rabbit monoclonal anti-PDI antiserum (Cell Signaling Technology); (5) rabbit anti-IRE1α monoclonal antibody (Cell Signaling Technology); (6) rabbit monoclonal anti-PERK antiserum (Cell Signaling Technology); (7) rabbit anti-phospho-IRE1α^Ser724^ polyclonal antibody (ThermoFisher); (8) rabbit monoclonal anti-phospho-PERK^Thr980^ antiserum (ThermoFisher); (9) rabbit anti-ATF4 polyclonal antibody (Proteintech, Rosemont, IL, USA); (10) mouse monoclonal anti-CHOP antiserum (Cell Signaling Technology); (11) rabbit anti-caspase-12 polyclonal antibody (Cell Signaling Technology); (12) rabbit monoclonal anti-Bim antiserum (Cell Signaling Technology); (13) rabbit anti-Noxa polyclonal antibody (Invitrogen); (14) rabbit monoclonal anti-Puma antiserum (Cell Signaling Technology). The membranes were then incubated with an HRP-linked secondary antibody, and the immunoreactive proteins were visualized with an ECL reagent (Merck).

### 4.11. Quantitative Real-Time RT-PCR Analysis

The total RNA of control or transfected dopaminergic neurons was extracted with Trizol Reagent (ThermoFisher). Then, the first-strand cDNA was produced using the total RNA (20 μg), SuperScript III RT (100 U, ThermoFisher), dNTP (1 mM), oligodT primer (8 ng/μL) and ribonuclease inhibitor (20 U) at 50 °C for 1 h. Subsequently, a StepOne Real-Time PCR system (Applied Biosystems, ThermoFisher, Waltham, MA, USA) with the SYBR Green PCR Master Mix (Applied Biosystems) was used to conduct a real-time PCR. The mRNA expression of GAPDH was quantified as the internal control. The relative change in the mRNA expression was determined with the following equation: Fold change = 2^−(ΔΔCt)^. ΔΔCt = (Ct_gene_ − Ct_GAPDH_)_shRNA_ − (Ct_gene_ − Ct_GAPDH_)_Con_.

### 4.12. Determination of Mitochondrial Superoxide and Mitochondrial Membrane Potential (ΔΨm)

To conduct live cell imaging of ΔΨm, control or transfected dopaminergic neurons seeded into 6-well plates were stained with ΔΨm-sensitive dye tetramethylrhodamine methyl ester (TMRM; 100 nM; ThermoFisher) for 30 min at 37 °C. Mitochondrial superoxide was visualized by treating dopaminergic neurons for 30 min at 37 °C with 5 μM MitoSox Red dye (ThermoFisher), which is oxidized by superoxide and generates red fluorescence. A LionHeart FX automatic microscope (BioTek) was used to obtain MitoSox Red or TMRM fluorescent images of 24 fields, which contain about 800–850 neurons, per well. Imaging signals of MitoSox Red or TMRM were examined with Gen5 software (BioTek), and the fluorescence intensity of transfected dopaminergic neurons was compared with that of control neurons.

### 4.13. Western Blot Assays of Mitochondria-Mediated Apoptotic Pathway

Cytosolic or mitochondrial protein samples of control or transfected dopaminergic neurons were separated on SDS-PAGE (15%) gels. PVDF membranes were then incubated with the following primary antibodies: (1) rabbit monoclonal anti-cleaved active caspase 9 antiserum (Cell Signaling Technology); (2) rabbit anti-cleaved active caspase 3 monoclonal antibody (Cell Signaling Technology); (3) mouse anti-cytochrome c monoclonal antibody (Abcam). Following the interaction with an HRP-linked secondary antiserum, the immunoreactive proteins of membrane were detected using ECL kit (Merck).

### 4.14. TUNEL Analysis of Dopaminergic Neurons

TUNEL staining was conducted with the In Situ Cell Death Detection Kit (Merck). Control or transfected dopaminergic neurons were fixed with paraformaldehyde and permeabilized with Triton X-100. Subsequently, dopaminergic neurons were interacted with the TUNEL reaction mixture for 1 h at 37 °C and then incubated with a Converter-peroxidase solution for 30 min at 37 °C. A DAB detection kit (Vector Labs, Newark, CA, USA) was used to visualize TUNEL-positive neurons.

### 4.15. Statistics

The results are indicated as the mean ± S.E. A one-way ANOVA with a following Tukey’s test (GraphPad Prism Program, version 9) was conducted to evaluate the statistical difference among multiple groups of data. An unpaired Student’s *t*-test (two-tailed) was used to examine the statistical difference between two groups of data. A *p* value of less than 0.01 was considered significant.

## 5. Conclusions

In summary, our data suggest that WT DNAJC6 supports the viability of dopaminergic neurons and that a PARK19 mutation-induced DNAJC6 deficiency impairs macroautophagy and decreases the protein expression of lysosomal protease cathepsin D by reducing the cytosolic protein level of clathrin heavy chain and the number of lysosomes. Further studies propose that a deficiency in the DNAJC6-induced downregulation of protease cathepsin D and macroautophagy impairment upregulates ER and mitochondrial levels of α-synuclein or phospho-α-synuclein^Ser129^ and activates ER stress-evoked and mitochondrial apoptotic signaling, resulting in the apoptotic death of dopaminergic neurons and PARK19. Our study provides the evidence that PARK19 homozygous mutations of DNAJC6 (Q789X or R927G) lead to the degeneration of dopaminergic neurons by diminishing the DNAJC6-mediated neuroprotective function. Our results further support the hypothesis that an upregulated expression of pathologic α-synuclein is a major molecular pathomechanisms underlying the degeneration of SN dopaminergic neurons observed in sporadic and hereditary PD cases [15,21,22]. A future in vivo study using a knock-in mouse model of PARK19 is required to demonstrate that PARK19 mutations of DNAJC6 cause the degeneration of SN dopaminergic neurons and resulting parkinsonism motor deficits by downregulating cathepsin D and upregulating pathologic α-synuclein.

## Figures and Tables

**Figure 1 ijms-25-06711-f001:**
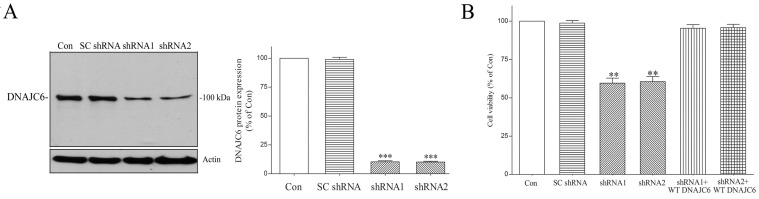
DNAJC6 paucity induces neurodegeneration of dopaminergic cells. (**A**) SC shRNA and shRNA of DNAJC6 were transfected to differentiated SH-SY5Y dopaminergic neurons for 3 days. Transfecting dopaminergic neurons with shRNA resulted in an approximately 90% decrease in DNAJC6 expression. SC shRNA failed to affect the DNAJC6 protein level. Each bar shows the mean ± S.E. value of 8 experiments. (**B**) DNAJC6 deficiency induced by a 4-day transfection of DNAJC6 shRNA led to significant cell death of dopaminergic neurons. Co-transfection of cDNA encoding WT human DNAJC6 prevented DNAJC6 shRNA-induced neurodegeneration of dopaminergic cells. Each bar represents the mean ± S.E. value of 8 experiments. ** *p* < 0.01 or *** *p* < 0.001 compared with control dopaminergic neurons.

**Figure 2 ijms-25-06711-f002:**
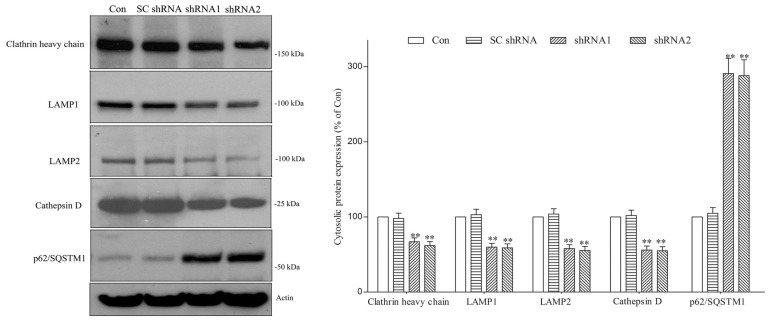
DNAJC6 deficiency causes downregulated expression of clathrin heavy chain, LAMP1, LAMP2 or cathepsin D and upregulated expression of p62/SQSTM1 in dopaminergic neurons. After transfection of shRNAs targeting DNAJC6 for 3 days, the protein level of cytosolic clathrin heavy chain was significantly decreased in dopaminergic neurons. Cytosolic protein levels of lysosomal marker proteins LAMP1 or LAMP2 and lysosomal aspartic protease cathepsin D were downregulated in DNAJC6 shRNA-transfected dopaminergic neurons. The protein level of cytosolic autophagy receptor p62/SQSTM1 was upregulated in dopaminergic neurons transfected with shRNAs targeting DNAJC6. Each bar shows the mean ± S.E. value of 6 experiments. ** *p* < 0.01 compared with control dopaminergic neurons.

**Figure 3 ijms-25-06711-f003:**
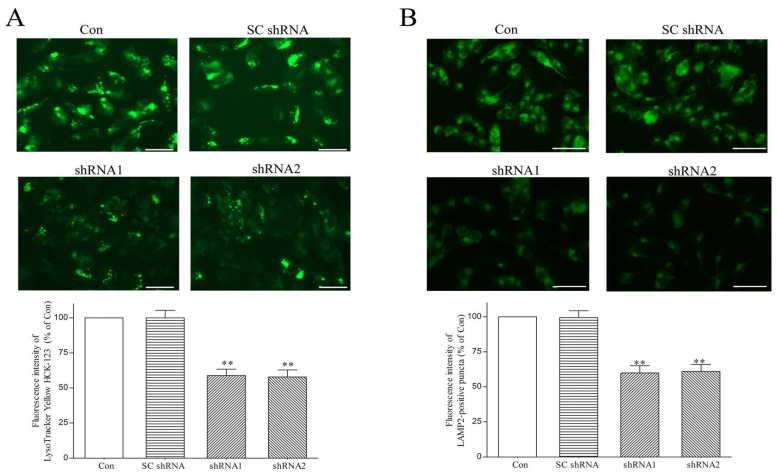
Paucity of DNAJC6 decreases LysoTracker staining intensity of lysosomes and immunofluorescence staining intensity of LAMP2-positive lysosomes in dopaminergic neurons. (**A**) Live cell imaging of lysosomes was conducted by treating differentiated SH-SY5Y dopaminergic neurons with 100 nM LysoTracker Yellow HCK-123, which stains acidic compartments and visualizes lysosomes, for 1 h. A three-day transfection of shRNA1 or shRNA2 targeting DNAJC6 significantly decreased the fluorescence intensity of LysoTracker Yellow staining in dopaminergic neurons. SC shRNA failed to affect lysosomal staining of LysoTracker Yellow HCK-123. Scale bar is 50 μm. (**B**) Immunofluorescence staining of lysosomal marker protein LAMP2 was performed to visualize lysosomes of dopaminergic neurons. Following a 3-day transfection of shRNAs targeting DNAJC6, the fluorescence intensity of LAMP2-positive puncta was significantly decreased in dopaminergic neurons. SC shRNA did not affect the immunofluorescence staining intensity of LAMP2. Scale bar is 60 μm. Each bar shows the mean ± S.E. value of 6 experiments. ** *p* < 0.01 compared with control dopaminergic neurons.

**Figure 4 ijms-25-06711-f004:**
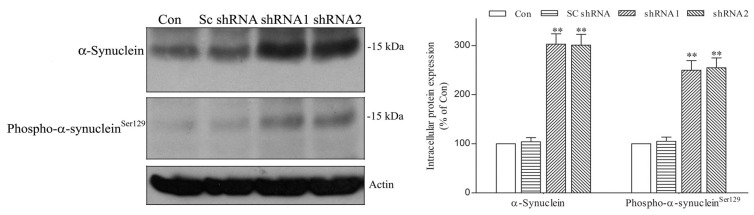
Deficiency of DNAJC6 elevates the protein level of intracellular α-synuclein or phospho-α-synuclein^Ser129^ in dopaminergic neurons. Immunoblotting analysis using cellular lysates showed that following a 3-day transfection of shRNAs targeting DNAJC6, the pathologic α-synuclein or phospho-α-synuclein^Ser129^ level was upregulated in dopaminergic neurons. Each bar represents the mean ± S.E. value of 6 experiments. ** *p* < 0.01 compared with control dopaminergic neurons.

**Figure 5 ijms-25-06711-f005:**
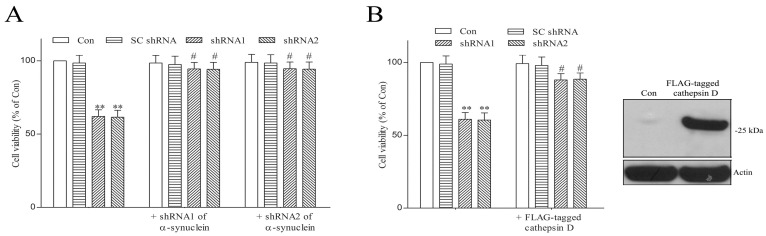
Transfection of α-synuclein shRNA or overexpression of FLAG-tagged cathepsin D blocks DNAJC6 deficiency-induced degeneration of dopaminergic neurons. (**A**) Co-transfecting dopaminergic neurons with shRNAs of α-synuclein significantly inhibited 4-day transfection of DNAJC6 shRNA-induced neurodegeneration. (**B**) FLAG-tagged cathepsin D was transiently overexpressed in SH-SY5Y dopaminergic neurons. FLAG-tagged cathepsin D significantly prevented DNAJC6 shRNA-evoked neuronal death of dopaminergic cells. Each bar shows the mean ± S.E. value of 6 experiments. ** *p* < 0.01 compared with control SH-SY5Y dopaminergic neurons. ^#^ *p* < 0.01 compared with DNAJC6 shRNA-transfected dopaminergic neurons.

**Figure 6 ijms-25-06711-f006:**
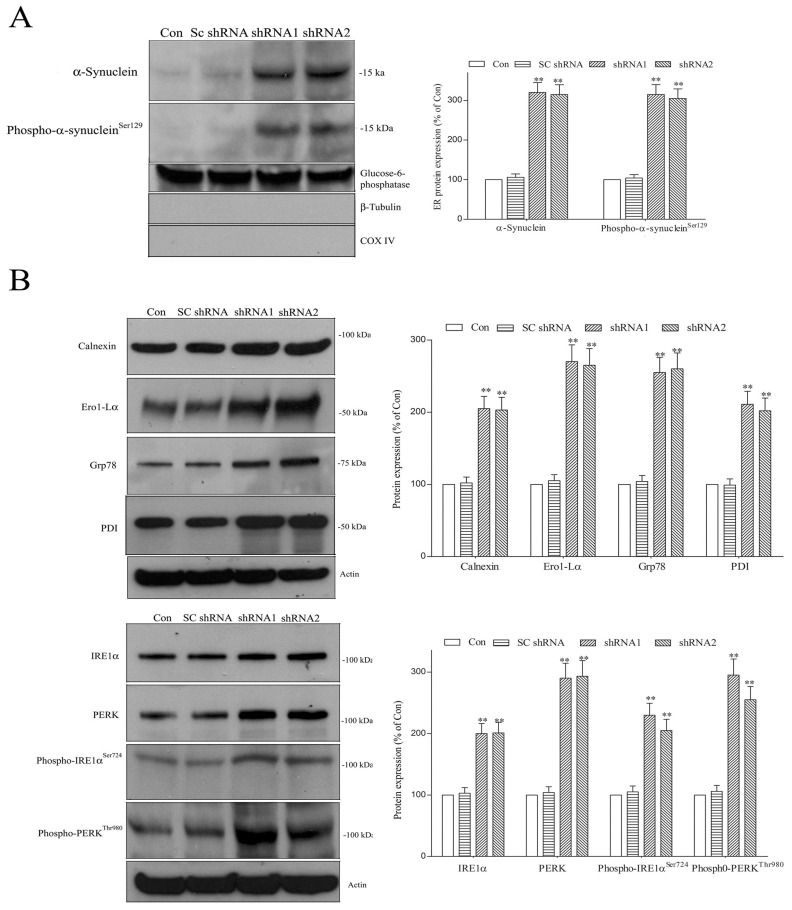
Paucity of DNAJC6 upregulates ER α-synuclein or phospho-α-synuclein^Ser129^ and activates ER stress and the UPR in dopaminergic neurons. (**A**) DNAJC6 deficiency caused by 3-day transfection of DNAJC6 shRNAs significantly increased ER α-synuclein or phospho-α -synuclein^Ser129^ in dopaminergic neurons. Glucose 6-phosphotase is an ER protein marker. Mitochondrial protein marker COX IV and cytoplasm protein marker β-tubulin were not expressed in the ER. (**B**) Three days after transfection of DNAJC6 shRNA, upregulated protein levels of ER chaperones, including calnexin, Ero1-Lα, Grp78 and PDI, and UPR markers, including IRE1α, PERK, phospho-IRE1α^Ser724^ and phospho-PERK^Thr980^, were observed in dopaminergic neurons. Each bar represents the mean ± S.E. value of 6 experiments. ** *p* < 0.01 compared with control dopaminergic neurons.

**Figure 7 ijms-25-06711-f007:**
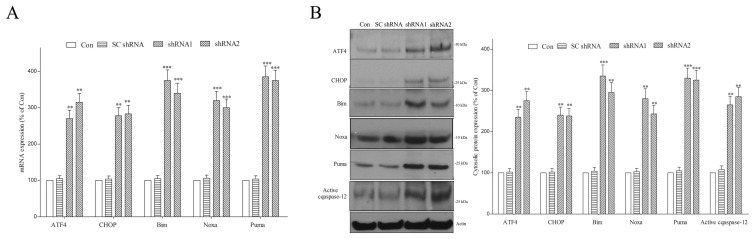
DNAJC6 paucity causes stimulation of ER stress-triggered apoptotic signaling in dopaminergic neurons. (**A**) Three-day transfection of shRNAs targeting DNAJC6 significantly increased mRNA expressions of ATF4, Bim, CHOP, Puma and Noxa in dopaminergic neurons. (**B**) DNAJC6 deficiency induced by 3-day transfection of DNAJC6 shRNAs upregulated protein levels of cytosolic active caspase-12, Puma, Noxa, Bim, CHOP and ATF4 in dopaminergic neurons. Each bar shows the mean ± S.E. value of 6 experiments. ** *p* < 0.01 or *** *p* < 0.001 compared with control dopaminergic neurons.

**Figure 8 ijms-25-06711-f008:**
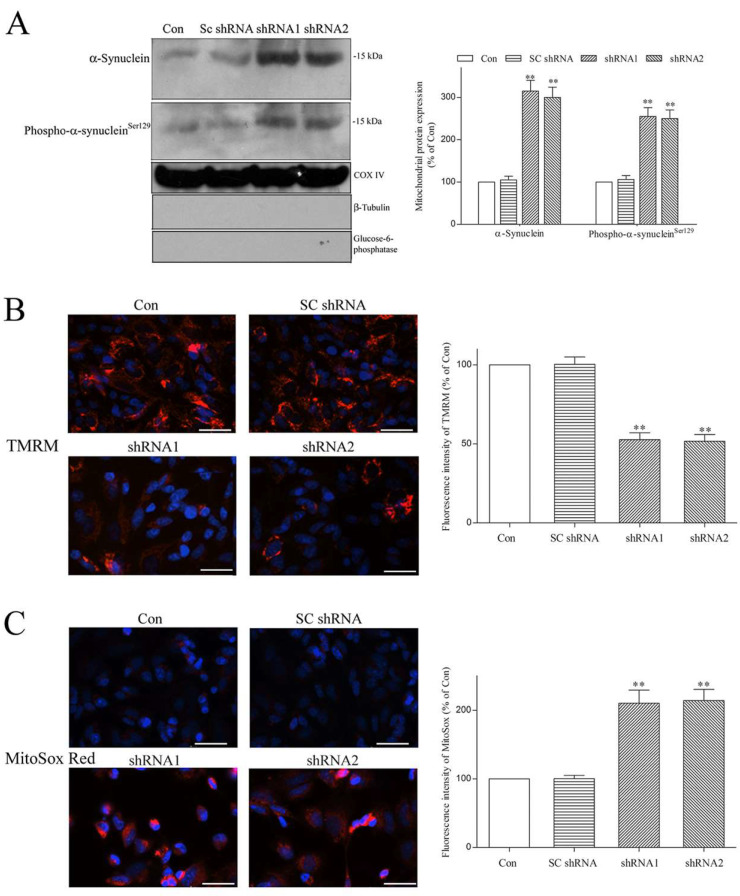
Lack of DNAJC6 upregulates α-synuclein or phospho-α-synuclein^Ser129^ within mitochondria and leads to mitochondrial dysfunction and oxidative stress in dopaminergic neurons. (**A**) DNAJC6 paucity caused by 3-day transfection of DNAJC6 shRNAs increased mitochondrial neurotoxic α-synuclein and phospho-α-synuclein^Ser129^. COX IV is a mitochondrial protein marker. ER protein marker glucose-6-phosphatase and cytoplasm protein marker β-tubulin were absent in the mitochondrial fraction. (**B**) Control or transfected dopaminergic neurons were treated with mitochondrial membrane potential (ΔΨm)-sensitive dye TMRM (100 nM) for 30 min at 37 °C. DNAJC6 shRNA-transfected dopaminergic neurons exhibited a reduced red fluorescence intensity of ΔΨm-sensitive TMRM and depolarized ΔΨm. Blue fluorescence indicates nuclear DAP1 staining. Scale bar is 50 μm. (**C**) Control or transfected dopaminergic neurons were incubated with 5 μM MitoSox Red dye for 30 min at 37 °C. Elevated fluorescence signals of MitoSox Red and mitochondrial superoxide were observed in dopaminergic neurons transfected with shRNAs targeting DNAJC6 for three days. Blue fluorescence shows nuclear DAP1 staining. Scale bar is 50 μm. Each bar shows the mean ± S.E. value of 6 experiments. ** *p* < 0.01 compared with control dopaminergic neurons.

**Figure 9 ijms-25-06711-f009:**
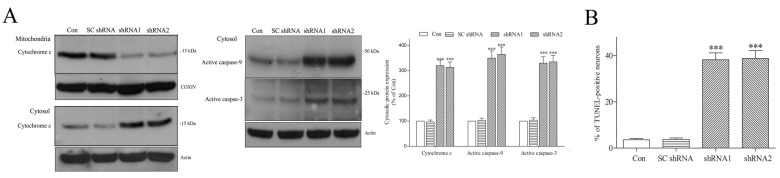
Deficiency of DNAJC6 activates mitochondria-mediated pro-apoptotic pathway in dopaminergic neurons. (**A**) Cytochrome c was predominantly located within mitochondrial fractions of control and SC shRNA-transfected dopaminergic neurons. Three-day transfection of shRNAs targeting DNAJC6 significantly decreased mitochondrial cytochrome c and elevated cytosolic cytochrome c in dopaminergic neurons. Transfecting dopaminergic neurons with shRNAs of DNAJC6 significantly upregulated cytosolic active caspase-9 and active caspase-3. (**B**) DNAJC6 paucity induced by 4-day transfection of shRNAs targeting DNAJC6 significantly upregulated the percentage of TUNEL-positive apoptotic dopaminergic neurons. Each bar represents the mean ± S.E. value of 6 experiments. *** *p* < 0.001 compared with control dopaminergic neurons.

**Figure 10 ijms-25-06711-f010:**
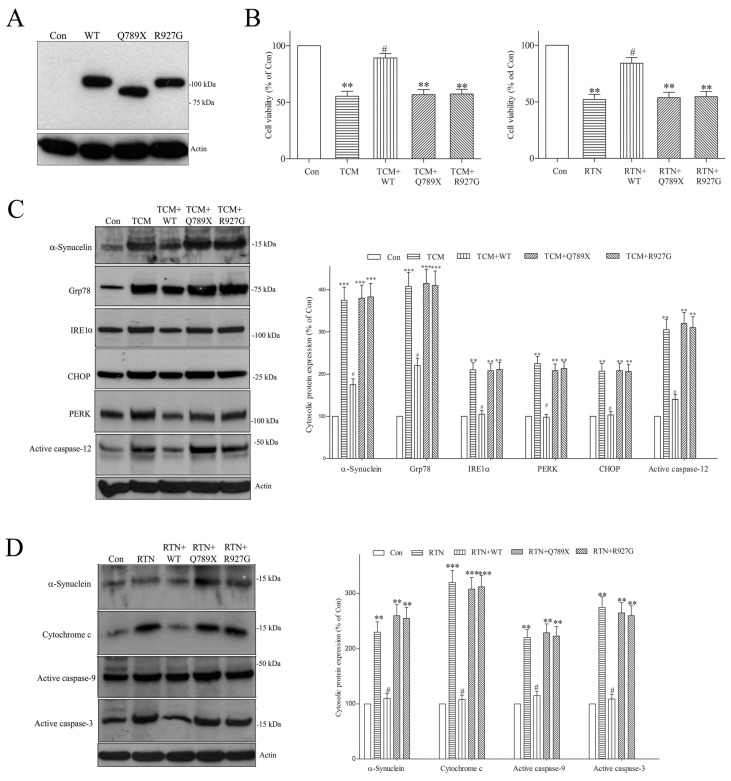
PARK19 DNAJC6 mutants fail to inhibit tunicamycin- or rotenone-evoked neurotoxicity, α-synuclein upregulation and apoptotic signaling in dopaminergic neurons. (**A**) FLAG-tagged human WT, Q789X or R927G DNAJC6 was transiently expressed in SH-SY5Y dopaminergic neurons. (**B**) Three-day incubation of 1 μM tunicamycin (TCM) or 0.5 μM rotenone (RTN) induced the neurodegeneration of dopaminergic cells. WT DNAJC6 significantly inhibited tunicamycin- or rotenone-evoked degeneration of dopaminergic neurons. On the contrary, PARK19 DNAJC6 mutants (Q789X or R927G) failed to prevent tunicamycin- or rotenone-triggered neurodegeneration. Each bar shows the mean ± S.E. value of 7 experiments. (**C**) Tunicamycin (TCM; 1 μM) upregulated the cytosolic expressions of active caspase-12, PERK, CHOP, IRE1α, Grp78 and α-synuclein. WT DNAJC6 significantly prevented the tunicamycin-evoked upregulation of cytosolic active caspase-12, PERK, CHOP, IRE1α, Grp78 and α-synuclein. PARK19 DNAJC6 mutants (Q789X or R927G) did not affect the upregulation of cytosolic active caspase-12, PERK, CHOP, IRE1α, Grp78 or α-synuclein caused by tunicamycin. (**D**) Rotenone (RTN; 0.5 μM) upregulated the cytosolic levels of α-synuclein, active caspase-3, active caspase-9 and cytochrome c. WT DNAJC6 inhibited the rotenone-evoked upregulation of cytosolic α-synuclein, active caspase-3, active caspase-9 and cytochrome c. PARK19 DNAJC6 mutants (Q789X or R927G) failed to reverse the upregulation of cytosolic α-synuclein, active caspase-3, active caspase-9 and cytochrome c caused by rotenone. Each bar represents the mean ± S.E. value of 6 experiments. ** *p* < 0.01 or *** *p* < 0.001 compared with control dopaminergic neurons. ^#^
*p* < 0.01 compared with tunicamycin- or rotenone-treated dopaminergic neurons.

## Data Availability

All data generated or analyzed during this study are included in this published article.

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
