# Peer review of "Downregulation of Protease Cathepsin D and Upregulation of Pathologic α-Synuclein Mediate Paucity of DNAJC6-Induced Degeneration of Dopaminergic Neurons"

_ijms, 2024, doi:10.3390/ijms25126711_

Round 1

Reviewer 1 Report

Comments and Suggestions for Authors

While the study shows that DNAJC6 deficiency leads to both ER stress and mitochondrial dysfunction, it does not thoroughly investigate the potential crosstalk between these two pathways, which could provide valuable insights into the pathogenesis of PARK19. Additionally, the study is limited in its examination of PARK19 mutants, focusing only on two variants (Q789X and R927G), and does not assess the functional consequences of dopaminergic neuron death caused by DNAJC6 deficiency. Expanding the analysis to include more PARK19 mutants and incorporating functional assays to measure the impact of DNAJC6 deficiency on dopaminergic neuron physiology would strengthen the study's findings and provide a more comprehensive understanding of the role of DNAJC6 in PARK19 pathology.

1.      The authors show that DNAJC6 deficiency leads to various cellular defects, but they do not demonstrate that restoring DNAJC6 expression can rescue these phenotypes.

2.       The study only examines two PARK19 mutants (Q789X and R927G).

3.       While the study suggests that cathepsin D downregulation contributes to α-synuclein accumulation, the direct link between these two events is not thoroughly explored.

4.       The study shows that DNAJC6 deficiency reduces the number of lysosomes but does not provide a detailed mechanism for how DNAJC6 regulates lysosomal homeostasis.

5.       study demonstrates that DNAJC6 deficiency leads to both ER stress and mitochondrial dysfunction but does not thoroughly investigate the potential crosstalk between these two pathways.

6.       While the study shows that DNAJC6 deficiency leads to the death of dopaminergic neurons, it does not assess the functional consequences of this loss.  

Author Response

The following are the reply to comments given by Referee #1 and the revisions I have made based on Reviewer #1’s comments:

  1. Referee #1 asked us to provide the results showing that restoring DNAJC6 expression rescues DNAJC6 deficiency-induced phenotypes in dopaminergic neurons. As shown in Figure 1B, shRNA of DNAJC6-induced DNAJC6 paucity caused degeneration of differentiated SH-SY5Y dopaminergic neurons. Restoring DNAJC6 expression caused by co-transfection of cDNA encoding WT human DNAJC6 prevented DNAJC6 shRNA-induced cell death of dopaminergic neurons (please see line 10, page 8 of Results section and Fig. 1B of revised manuscript).
  2. Reviewer #1 mentioned that we only examined two PARK19 DNAJC6 mutants, (Q789X) and (R927G) DNAJC6, in the original manuscript and suggested that we should also investigate functional property of other PARK19 DNAJC6 mutants. Previous study identified only one homozygous missense mutation of DNAJC6, (R927G) mutation, as the cause of PARK19. Several homozygous nonsense truncating mutations of DNAJC6, including (R256X), (Q734X), (Q789X) and (R806X) mutations, are the genetic cause of PARK19. In this study, we examined neuroprotective function of PARK19 mutant (Q789X) or (R927G) DNAJC6 and tested the hypothesis that PARK19 mutation of DNAJC6 causes the loss of DNAJC6-induced neuroprotective effect and resulting neurodegeneration of dopaminergic cells. Our results indicate that in contrast with neuroprotective function of WT DNAJC6, PARK19 mutant (Q789X) or (R927G) DNAJC6 fails to suppress tunicamycin- or rotenone-evoked upregulation of pathologic a-synuclein and stimulation of apoptotic pathway within dopaminergic neurons (Fig. 10). Compared to mutant (R256X) or (Q734X) DNAJC6, mutant (Q789X) DNAJC6 has more amino acid residues. Mutant (Q789X) and (R806X) DNAJC6 have a similar number of amino acid residues. As a result, it is almost certain that similar to (Q789X) DNAJC6 mutation, truncating (R256X), (Q734X) or (R806X) mutation also causes the loss of DNAJC6-mediated neuroprotective effect and subsequent degeneration of dopaminergic neurons. Therefore, it is not necessary for us to examine functional property of mutant (R256X), (Q734X) or (R806X) DNAJC6.
  3. Referee #1 asked us to provide the results showing the direct link between downregulation of cathepsin D and accumulation of a-synuclein observed in our study. Cathepsin D is a major lysosomal aspartic protease and mainly responsible for macroautophagy or CMA-mediated lysosomal degradation of a-synuclein. We hypothesized that DNAJC6 deficiency-induced reduction of lysosomal number (Fig. 3) could decrease protein level of lysosomal protease cathepsin D within dopaminergic neurons and cause resulting upregulation of neurotoxic a-synuclein. In accordance with our hypothesis, transfection of DNAJC6 shRNAs significantly reduced cytosolic level of cathepsin D (Fig. 2) and increased intraneuronal a-synuclein or phospho-a-synucleinSer129 level in dopaminergic neurons (Fig. 4). To provide the direct evidence that DNAJC6 paucity-induced downregulation of protease cathepsin D causes upregulation of neurotoxic a-synuclein and resulting degeneration of dopaminergic neurons. FALG-tagged cathepsin D was overexpressed in DNAJC6 shRNA-transfected dopaminergic neurons (Fig. 5B). Following overexpression of FLAG-tagged cathepsin D, shRNA1 or shRNA2 of DNAJC6 did not significantly increase protein level of intraneuronal a-synuclein (n = 4 experiments) (please see line 1, page 11 of Results section). Overexpression of FALG-tagged cathepsin D also prevented shRNA of DNAJC6-induced degeneration of dopaminergic neurons (Fig. 5B). In my opinion, these results prove that there is a direct link between downregulation of cathepsin D and resulting accumulation of pathologic a-synuclein, which then causes cell death of dopaminergic neurons.
  4. Reviewer #1 asked us to provide a detailed mechanism by which DNAJC6 deficiency reduces the number of lysosomes in dopaminergic neurons. As mentioned in Introduction section of original manuscript, normal intracellular level of free clathrin is required for the procedure of autophagic lysosome reformation (ALR), which is needed for maintaining the number and homeostasis of functional lysosomes. DNAJC6 plays a critical role in regulating homeostasis of intracellular free clathrin by mediating clathrin uncoating of clathrin-coated vesicles, which is an essential step during clathrin-mediated endocytosis DNAJC6 deficiency-induced defective clathrin-uncoating process could cause a reduction in intraneuronal level of free clathrin and impairment of ALR, leading to dysregulated lysosomal homeostasis and reduced number of lysosomes. Consistent with our hypothesis, shRNA of DNAJC6-induced DNAJC6 deficiency significantly decreased cytosolic protein level of clathrin heavy chain in dopaminergic neurons (Fig. 2). Therefore, DNAJC6 deficiency decreases the number of lysosomes in dopaminergic neurons by downregulating cytosolic level of clathrin heavy chain and impairing the function of ALR (please see line 10, page 16 of Discussion section).
  5. The results of this manuscript demonstrate that DNAJC6 deficiency causes both activation of ER stress and mitochondrial dysfunction. Referee #1 asked whether there is a potential crosstalk between these two pathways. As shown in Fig. 6, DNAJC6 deficiency upregulated protein level of ER a-synuclein or phospho-a-synucleinSer129 in dopaminergic neurons, leading to activation of ER stress, which is shown by upregulation of calnexin, Ero1-La, Grp78 or PDI, and UPR, which is shown by upregulation of IRE1a, PERK, phospho-IRE1aSer724 or phospho-PERKThr980. In the presence of incessant ER stress, UPR induces activation of ER stress-evoked apoptotic signaling, which is shown by upregulated mRNA and protein expression of ATF4, Bim, CHOP, Puma or Noxa (Fig. 7). As shown in Fig. 8, paucity of DNAJC6 also increased mitochondrial level of a-synuclein or phospho-a-synucleinSer129, leading to depolarization of mitochondrial membrane potential (ΔΨm) and elevation of mitochondrial ROS. Upregulation of pro-apoptotic Bim, Puma or Noxa stimulates mitochondria-mediated apoptotic pathway via enhancing translocation of mitochondrial cytochrome c to cytoplasm and activating caspase-9 and caspase-3. Upregulation of mitochondrial ROS-induced oxidative stress and depolarization of Ψm-induced mitochondrial dysfunction also triggers activation of mitochondrial pro-apoptotic signaling by augmenting release of mitochondrial cytochrome c. Therefore, DNAJC6 deficiency-induced two pathways, activation of ER stress and mitochondrial dysfunction, converge on the activation of mitochondria-mediated pro-apoptotic signaling (Fig. 9A) and causing apoptotic death of dopaminergic neurons (Fig. 9B). As a result, there is a crosstalk between DNAJC6 deficiency-evoked ER stress and DNAJC6 paucity-induced mitochondrial dysfunction.
  6. The results of this manuscript suggest that deficiency of DNAJC6-induced downregulation of protease cathepsin D and macroautophagy impairment upregulates ER and mitochondrial levels of a-synuclein or phospho-a-synucleinSer129 and activates ER stress-evoked and mitochondrial apoptotic signaling, resulting in apoptotic death of dopaminergic neurons. Reviewer #1 mentioned that although our study sheds a light on molecular mechanisms of DNAJC6 deficiency-induced loss of dopaminergic neurons, but our study does not assess the functional consequences of this loss. According to results of our in vitro study, we can hypothesize that PARK19 mutation-induced DNAJC6 deficiency upregulates neurotoxic a-synuclein or phospho-a-synucleinSer129 of substantia nigra (SN) dopaminergic neurons and causes apoptotic death of SN dopaminergic cells, leading to pathological consequence, which is shown by the presence of parkinsonism motor deficits. As mentioned in Conclusion section (line 14, page 26) of revised manuscript, future in vivo study using knockin mouse model of PARK19 is required to demonstrate that PARK19 mutation of DNAJC6 causes degeneration of SN dopaminergic neurons and resulting parkinsonism motor deficits by downregulating cathepsin D and upregulating pathologic a-synuclein. Currently, we are investing molecular pathogenic mechanism of PARK19 by using homozygous Dnajc6Q787X/Q787X knockin mouse, which is animal model of PARK19.

Reviewer 2 Report

Comments and Suggestions for Authors

Comments and suggestions for each section:

Introduction

  • The flow of the introduction could be more structured. The authors discuss various aspects of PD, including the role of α-synuclein, autophagy-lysosome pathways, and ER stress, but the transitions between these topics could be smoother to improve readability.
  • The hypothesis and objectives of the study are mentioned only at the end of the introduction. It would be better to state the hypothesis and objectives earlier to provide a clear direction for the reader.
  • The authors mention several DNAJC6 mutations associated with PARK19, but they do not clearly explain how these mutations lead to a loss of function of the protein. Providing a brief explanation of the functional consequences of these mutations would strengthen the introduction.
  • The introduction could benefit from a more concise summary of the key points and a clearer statement of the knowledge gap that this study aims to address.
  • The authors could consider using subheadings to organize the different sections of the introduction, such as "DNAJC6 mutations in PARK19," "The role of α-synuclein in PD," "Autophagy-lysosome pathways and PD," and "ER stress and mitochondrial dysfunction in PD."

Results

  • The organization of the results could be more structured. The authors present their findings in a logical order, but the use of subheadings for each major result would improve readability and help the reader navigate through the different aspects of the study.
  • Some of the figure legends lack sufficient detail. For example, in Figure 3, the authors should specify the duration of LysoTracker Yellow HCK-123 treatment and the concentration used. Similarly, in Figure 8B and 8C, the concentrations of TMRM and MitoSox Red should be mentioned.
  • The authors should provide more information about the statistical analyses performed. They mention using mean ± S.E. and indicate statistical significance using asterisks, but they do not specify the statistical tests used or the number of replicates for each experiment.
  • The results section could benefit from a brief discussion of the limitations of the study. For example, the authors use a cellular model of DNAJC6 deficiency, which may not fully recapitulate the complex pathophysiology of PARK19.
  • The authors demonstrate that PARK19 mutant DNAJC6 fails to protect against tunicamycin- or rotenone-induced neurotoxicity, which is an important finding. However, they do not provide a clear mechanistic explanation for why these mutants lack neuroprotective function.

Discussion

  • The authors could provide a more detailed comparison of their findings with those of previous studies on DNAJC6 and its role in PD. This would help to highlight the novelty and significance of their work.
  • The discussion of the potential therapeutic implications of the study is limited. The authors could expand on how their findings might inform the development of new strategies for the treatment of PARK19 and other forms of PD.
  • The authors could discuss the limitations of their study in more detail. For example, they could mention that the use of a cellular model may not fully recapitulate the complex pathophysiology of PD and that further in vivo studies are needed to validate their findings.
  • The authors could provide a more detailed discussion of the potential mechanisms by which PARK19 mutant DNAJC6 fails to protect against tunicamycin- or rotenone-induced neurotoxicity. This would help to strengthen the link between their findings and the pathogenesis of PARK19.
  • The discussion could benefit from a more concise conclusion that summarizes the main findings of the study and their potential implications for future research and clinical practice.

Methods

  • The authors should provide more information about the validation of the shRNAs used to knock down DNAJC6 and α-synuclein expression. They should report the knockdown efficiency and any potential off-target effects.
  • The description of the construction of PARK19 mutant DNAJC6 cDNA is brief. The authors should provide more details about the specific mutations introduced and the primers used for site-directed mutagenesis.
  • The authors should include information about the number of replicates performed for each experiment and whether the experiments were repeated independently.
  • The statistical analysis section could be more detailed. The authors should specify the software used for the analysis and provide more information about the post-hoc tests performed.

Conclusion

  • Discussing the potential limitations of the study and how they might impact the interpretation of the results.
  • Providing a brief outlook on future research directions, such as the need for in vivo studies to validate the findings and the potential for targeting the identified molecular pathways for therapeutic intervention.
  • Placing the findings in the broader context of Parkinson's disease research and discussing how they contribute to our understanding of the disease pathogenesis.

Author Response

The following are the reply to comments given by Referee #2 and the revisions I have made based on Reviewer #2’s comments:

  1. Various aspects of PD, including the role of α-synuclein, autophagy-lysosome pathways, autophagic lysosome reformation, cathespsin D, ER stress and mitochondrial dysfunction, were mentioned in Introduction As required by Referee #2, Introduction section has been revised to make the transition between these topics smoother.
  2. Reviewer #2 mentioned that we should state hypotheses and objectives of this study earlier in Introduction The objectives of this study were stated in the first part of Introduction section (please see line 19, page 4). Our hypotheses were also stated in initial parts of Introduction section (please see line 8, page 6; line 21, page 6 and line 7, page 7).
  3. Homozygous nonsense truncating (R256X), (Q734X), (Q789X) or (R806X) mutation of DNAJC6 and missense (R927G) mutation of DNAJC6 cause PARK19. It is generally believed that truncating or missense mutation of DNAJC6, which causes absence of full-length functional DNAJC6, leading to loss of DNAJC6-mediated neuroprotective effect in dopaminergic neurons (line 15, page 4 of Introduction section). Referee #2 asked how these mutations lead to a loss of DNAJC6 function. As described in Discussion section (line 15, page 19), J-domain of human DNAJC6, which is required for interacting with HSC70 and HSC70-DNAJC6 complex-mediated various physiological functions, is located at the C-terminal domain containing (R927) residue. As a result, PARK19 C-terminal truncating mutations and missense (R927G) mutation of DNAJC6 lead to a loss of DNAJC6 function. Reviewer #2 asked us to provide a brief explanation of the functional consequences of these mutations. As mentioned in Introduction section (line 15, page 4), we hypothesized that homozygous (Q789X) or (R927G) mutation of DNAJC6 causes loss of neuroprotective effect, resulting in degeneration of SN dopaminergic neurons and PARK19. Consistent with our hypothesis, the results of this manuscript, for the first time, provide the evidence that PARK19 homozygous (Q789X) or (R927G) mutation of DNAJC6 causes degeneration of dopaminergic neurons by impairing DNAJC6-mediated neuroprotective function (Fig. 10).
  4. Referee #2 mentioned that Introduction section should provide a concise summary of key results observed in this study and a clearer statement of research purpose. As mentioned in the first part of Introduction section (line 19, page 4), purposes of this study are to elucidate molecular pathomechanisms underlying DNAJC6 paucity-induced neurodegeneration of dopaminergic cells and to prove that PARK19 truncating or missense mutation of DNAJC6 impairs neuroprotective function of DNAJC6. Key results of this manuscript are also mentioned in the last part of Introduction section (line 16, page 7).
  5. As required by Reviewer #2, subheadings for each major result are used for Results.
  6. As instructed by Referee #2, the concentration and treatment duration of LysoTracker Yellow HCK-123 are mentioned in the figure legend of Figure 3. The concentrations of TMRM and MitoSox Red are also indicated in the figure legend of Figure 8.
  7. Reviewer #2 asked us to provide more information about statistical analyses performed. As mentioned in Methods section (line 5, page 25), one-way ANOVA analysis with subsequent Tukey’s test was conducted to determine statistical difference among multiple groups of data. Unpaired Student’s t-test (two-tailed) was used to evaluate statistical difference between two groups of data. The number of independent experiments performed for the results of each figure is also indicated in the figure legend.
  8. In this study, we studied molecular pathomechanisms underlying DNAJC6 deficiency-induced degeneration of dopaminergic neurons by using PARK19 cellular model, which may not fully recapitulate the complex pathophysiology of PARK19. As mentioned by Referee #2, future study using PARK19 mouse model is required to provide in vivo evidence that PARK19 mutation-induced DNAJC6 deficiency causes degeneration of SN dopaminergic neurons via downregulating protease cathepsin D and upregulating neurotoxic a-synuclein (please see line 3, page 11 of Results section).
  9. Reviewer #2 asked us to explain why PARK19 mutant (Q789X) or (R927G) DNAJC6 fails to attenuate tunicamycin- or rotenone-induced neurotoxicity and lacks neuroprotective function (Fig. 10). As mentioned in Discussion section (line 15, page 19), J-domain of human DNAJC6, which is required for interacting with HSC70 and HSC70-DNAJC6 complex-mediated physiological effects including neuroprotective function, is located at the C-terminal domain containing (R927) residue. As a result, mutant (Q789X) DNAJC6, which lacks C-terminal J-domain, and mutant (R927G) DNAJC6, which possesses loss-of-function (R927G) mutation of J-domain, fail to interact with HSC70 and produce neuroprotective effects.
  10. Referee #2 mentioned that we should compare the results of our study with previous studies on pathogenic mechanisms of mutant DNAJC6-induced PARK19 in Discussion However, the results of this manuscript, for the first time, shed a light on detailed molecular pathomechanisms by which PARK19 mutation-induced DNAJC6 deficiency causes degeneration of dopaminergic neurons and resulting PD.
  11. Reviewer #2 asked us to discuss potential therapeutic implications of this study. The results of this manuscript suggest that PARK19 mutation-induced DNAJC6 deficiency causes macroautophagy impairment and downregulation of lysosomal protease cathepsin D, leading to upregulated level of neurotoxic a-synuclein or phospho-a-synucleinSer129 and resulting neurodegeneration of SN dopaminergic cells. According to our results, it is reasonable to hypothesize that drugs, which activate macroautophagy or cathepsin D, could possess therapeutic effects on PARK19 degeneration of SN dopaminergic neurons (please line 17, page 17 of Discussion section).
  12. As requested by Reviewer #2, we discussed the potential mechanism by which PARK19 mutant (Q789X) or (R927G) DNAJC6 fails to protect against tunicamycin- or rotenone-induced neurotoxicity (please line 15, page 19 of Discussion section).
  13. As required by Referee #2, major findings or clinical implication of this study and potential future study are mentioned in Discussion section (line17, page 17) or Conclusion section (line 2, page 26).
  14. Reviewer #2 asked the knockdown efficiency of DNAJC6 shRNAs or a-synuclein shRNAs used in this manuscript. As indicated in Fig. 1 of Results section (line 7, page 8), transfection of human DNAJC6 shRNAs caused an about 90% decrease in protein expression of endogenous DNAJC6 (Fig. 1A). a-Synuclein shRNAs knocked down about 85 % of endogenous a-synuclein (line 17, page 10 of Results section). The specificity of DNAJC6 shRNA or a-synuclein shRNA was indicated by the finding that SC shRNAs failed to affect protein level of endogenous DNAJC6 (Fig. 1A) or a-synuclein (data not shown).
  15. As requested by Referee #2, mutation primers used for the preparation of cDNAs encoding mutant (Q789X) DNAJC6 and (R927G) DNAJC6 were indicated in Methods section (please see line 17, page 20).
  16. Reviewer #2 asked the number of independent experiments performed for the results of this study. The number of independent experiments conducted for the results of each figure is indicated in the figure legend of each figure.
  17. As required by Referee #2, detailed statistical analyses and software used for the analysis are indicated in Methods section (please see line 5, page 25).
  18. As mentioned by Reviewer #2, PARK19 cellular model used in this manuscript has limitation. Therefore, future in vivo study using knockin mouse model of PARK19 is required to demonstrate that PARK19 mutation of DNAJC6 causes degeneration of SN dopaminergic neurons and resulting parkinsonism motor deficits by downregulating cathepsin D and upregulating pathologic a-synuclein (please see line 14, page 26 of Conclusion section).
  19. Referee #2 asked us to discuss the contribution of our study to understanding the pathogenesis of Parkinson’s disease. The results of this manuscript propose that deficiency of DNAJC6-induced downregulation of protease cathepsin D and macroautophagy impairment upregulates ER and mitochondrial levels of a-synuclein or phospho-a-synucleinSer129 and activates ER stress-evoked and mitochondrial apoptotic signaling, resulting in apoptotic death of dopaminergic neurons and PARK19. Our results further support the hypothesis that upregulated expression of pathologic a-synuclein is a major molecular pathomechanisms underlying degeneration of SN dopaminergic neurons observed in sporadic and hereditary PD cases (please see line 11, page 26 of Conclusion section). .

Round 2

Reviewer 2 Report

Comments and Suggestions for Authors

The authors responded to my comments sufficiently.